# Healthy ageing of cloned sheep

K.D. Sinclair[1], S.A. Corr[1], C.G. Gutierrez[1,2], P.A. Fisher[1], J.-H. Lee[1,†], A.J. Rathbone[1], I. Choi[1,†], K.H.S. Campbell[1,✠] & D.S. Gardner[1]

The health of cloned animals generated by somatic-cell nuclear transfer (SCNT) has been of concern since its inception; however, there are no detailed assessments of late-onset, non-communicable diseases. Here we report that SCNT has no obvious detrimental long-term health effects in a cohort of 13 cloned sheep. We perform musculoskeletal assessments, metabolic tests and blood pressure measurements in 13 aged (7–9 years old) cloned sheep, including four derived from the cell line that gave rise to Dolly. We also perform radiological examinations of all main joints, including the knees, the joint most affected by osteoarthritis in Dolly, and compare all health parameters to groups of 5- and 6-year-old sheep, and published reference ranges. Despite their advanced age, these clones are euglycaemic, insulin sensitive and normotensive. Importantly, we observe no clinical signs of degenerative joint disease apart from mild, or in one case moderate, osteoarthritis in some animals. Our study is the first to assess the long-term health outcomes of SCNT in large animals.

[1] Schools of Biosciences (KDS, CGG, PAF, J-HL, AJR, IC, KHSC), School of Veterinary Medicine and Science (SAC, DSG), University of Nottingham, Leicestershire LE12 5RD, UK. [2] Universidad Nacional Autonoma de Mexico, Facultad de Medicina Veterinaria, Mexico City 04510, Mexico. † Present address: Department of Animal Bioscience, Institute of Agriculture & Life Science, College of Agriculture & Life Sciences, Gyeongsang National University, Jinju 52828, Republic of Korea (J.-H.L.); Division of Animal and Dairy Sciences, College of Agriculture and Life Sciences, Chungnam National University, Daejeon 34134, Republic of Korea (I.C.). Correspondence and requests for materials should be addressed to K.D.S. (email: kevin.sinclair@nottingham.ac.uk).
✠Deceased

The 20th anniversary of the birth of the first animal ('Dolly') to be derived from adult cells[1], celebrated in July 2016, marks a milestone in the progress of somatic-cell nuclear transfer (SCNT), which has since been undertaken in more than 20 mammalian species including, quite recently, work involving human cells[2–4]. The pioneering study of Wilmut et al.[1] was significant, however, because it demonstrated the potential to induce pluripotency in terminally differentiated cells. Indeed, although alternative approaches now exist for the generation of pluripotent stem cells[5], it is generally recognized that SCNT remains the most effective way to reprogram somatic cells to a pluripotent state[6,7]. Nevertheless, despite technological advances in recent years[8] efficiency of SCNT remains low. This is best exemplified by the well-documented losses of partially reprogrammed embryos occurring throughout gestation and immediately following birth[9–12]. Although detailed clinical studies are lacking, the general consensus is that cloned offspring that survive beyond the neonatal period are healthy and reproductively fit[13–17]; although in some studies cloned mice tended to be obese, hyperinsulinaemic[18] and were shorter lived[19].

Longevity and healthy ageing among SCNT clones have been contentious issues from the outset[20,21], with much made of the fact that Dolly had been undergoing treatment for osteoarthritis (OA) for some time before her death in 2003, aged 6½ years[22–24]. However, to the best of our knowledge, no formal, comprehensive assessment has been conducted of age-related, non-communicable diseases such as metabolic syndrome, hypertension and OA (three major comorbidities in aged human populations[25]) in cloned offspring of any species. This is significant because it is now well established that relatively subtle alterations to the periconceptional environment, either during natural mating or following assisted reproduction, can lead to heritable alterations to the epigenome linked to subsequent development and late-onset chronic diseases[26–29]. Given that SCNT requires the use of assisted reproductive procedures to generate donor oocytes and to culture reconstructed zygotes to the point of transfer, it is important to establish whether similar comorbidities exist in apparently healthy aged cloned offspring, using a long-lived and precocial mammalian species.

To this end we recently undertook detailed assessments of glucose tolerance, insulin sensitivity, ambulatory- and stimulated-blood pressure, together with musculoskeletal investigations, including clinical and radiological examination of all main joints, in 13 cloned sheep aged between 7 and 9 years (natural lifespan typically < 10 years). In addition, magnetic resonance imaging (MRI) was undertaken of the stifle joints (the joint between the femur and tibia). These animals originated from studies undertaken between 2005 and 2007 that sought to improve the efficiency of SCNT by addressing issues related to mitochondrial heteroplasmy and chromatin remodelling at the point of nuclear transfer. Pregnancy outcomes for some of the clones derived from primary fetal fibroblasts were reported previously[30,31]. In addition to reporting the health status of aged clones derived from those studies we also report on the health of four 8-year-old Finn-Dorset clones (Fig. 1) derived from the mammary gland cell line that gave rise to Dolly[1]. That is they are clones (genomic copies) of Dolly. Reference is made to the health status of contemporary groups of 5-and 6-year-old sheep, and published values for specific health-related parameters in sheep. Despite their advanced age, the clones were found to be euglycaemic, insulin sensitive and normotensive. Although most showed radiographic evidence of mild osteoathritis in one or two joints, no animal was lame and none have required treatment. We could find no evidence, therefore, of a detrimental long-term effect of cloning by SCNT on the health of aged offspring among our cohort.

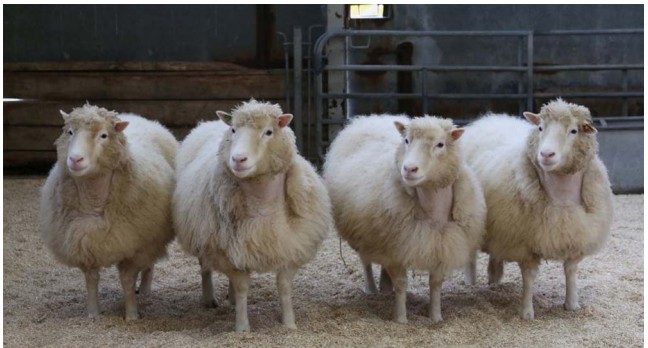

**Figure 1 | Four 8-year old Finn-Dorset clones born in July 2007 and derived from the mammary-gland cell line that gave rise to Dolly.**

## Results

**Origin and genetic identity of cloned offspring.** Microsatellite analyses confirmed that sheep within each clonal group were genetically identical to their respective parental-donor cell line for all 21 markers (Supplementary Table 1). The oldest clone (A089; male) was born in July 2006 and was derived from the primary fetal-fibroblast cell line SFF1 (Supplementary Table 1) as reported previously[31]. As part of that study the donor cell that gave rise to A089 had undergone mtDNA depletion before NT. Six of the female Lleyn clones were born in June 2008 and were derived from the primary fetal-fibroblast cell line LFF4 as reported previously[30]. Four of these clones (that is, 382, 385, 386 and 387) were derived from permeabilized fetal fibroblasts incubated with Xenopus oocyte extract with a view to enhancing epigenetic remodelling of chromatin before NT. The remaining two clones (that is, 388 and 389) were derived from permeabilized fibroblasts that were not exposed to extract. A seventh female Lleyn clone (2265) was also derived from LFF4 cells but was born in the previous year (August 2007) along with clone 2264 (derived from fetal-fibroblast cell line SFF5, breed unknown). These two clones formed part of a pilot study designed to assess the suitability of these two cell lines for SCNT using standard in-house protocols[30,31].

The four female Finn-Dorset clones (2260, 2261, 2262 and 2263; Fig. 1) were born in July 2007. Data from these animals have not been reported previously. They were derived from the mammary gland cell line that gave rise to Dolly[1] (Supplementary Table 1). The study that produced these four ewes (Supplementary Table 6) was an extension of that of Choi et al[32]. which used fetal fibroblasts as nuclear donors. Both studies investigated the effects of treating ovine oocytes during the latter stages of in vitro maturation, before nuclear transfer, with caffeine. We now report that 10 lambs derived from these nuclear donor cells were born (derived from seven caffeine-treated and three control oocytes), seven lived beyond 1 week of age (derived from five caffeine-treated and two control oocytes), but only four survived to adulthood (all derived from caffeine treated oocytes).

**Body composition and peripheral insulin sensitivity.** In preparation for metabolic assessments animals were fed a nutritionally-balanced diet of hay and concentrates ((metabolizable energy = 11.0 MJ kg$^{-1}$ DM; crude protein = 130 g kg$^{-1}$ DM) ad libitum (equivalent to ∼2 × maintenance[33])) for a 28-week period to attain a body condition score (BCS) of 4.0 units ($1_{[thin]}$ to $5_{[obese]}$ scale[34]), equivalent to ∼40% total body fat[35] for female sheep. This was in line with our reference population of 6-year-old females conceived by embryo transfer (ET-controls). Mean

(±s.e.m.) BCS of 4.0±0.1 and 4.1±0.2 units were attained for Lleyn and Finn-Dorset clones at the time of body composition assessment by dual-energy X-ray absorptiometry (DEXA). Body condition scores for clones 2264, 2265 and A089 at this time were 3.8, 3.8 and 3.0 units, respectively. Subsequent body fat content determined by DEXA was slightly greater (analysis of variance

(ANOVA); $P<0.01$) in Finn-Dorset clones ($\sim46\%$) than ET-Controls ($\sim40\%$), but was lower ($\sim21\%$; $P<0.001$) in Lleyn clones (Fig. 2a). This was despite the fact that they were all fed the same diet *ad libitum* for an extended period (28 weeks), and BCS were similar. This probably reflects differences in body fat (that is, subcutaneous versus intra-abdominal fat) distribution

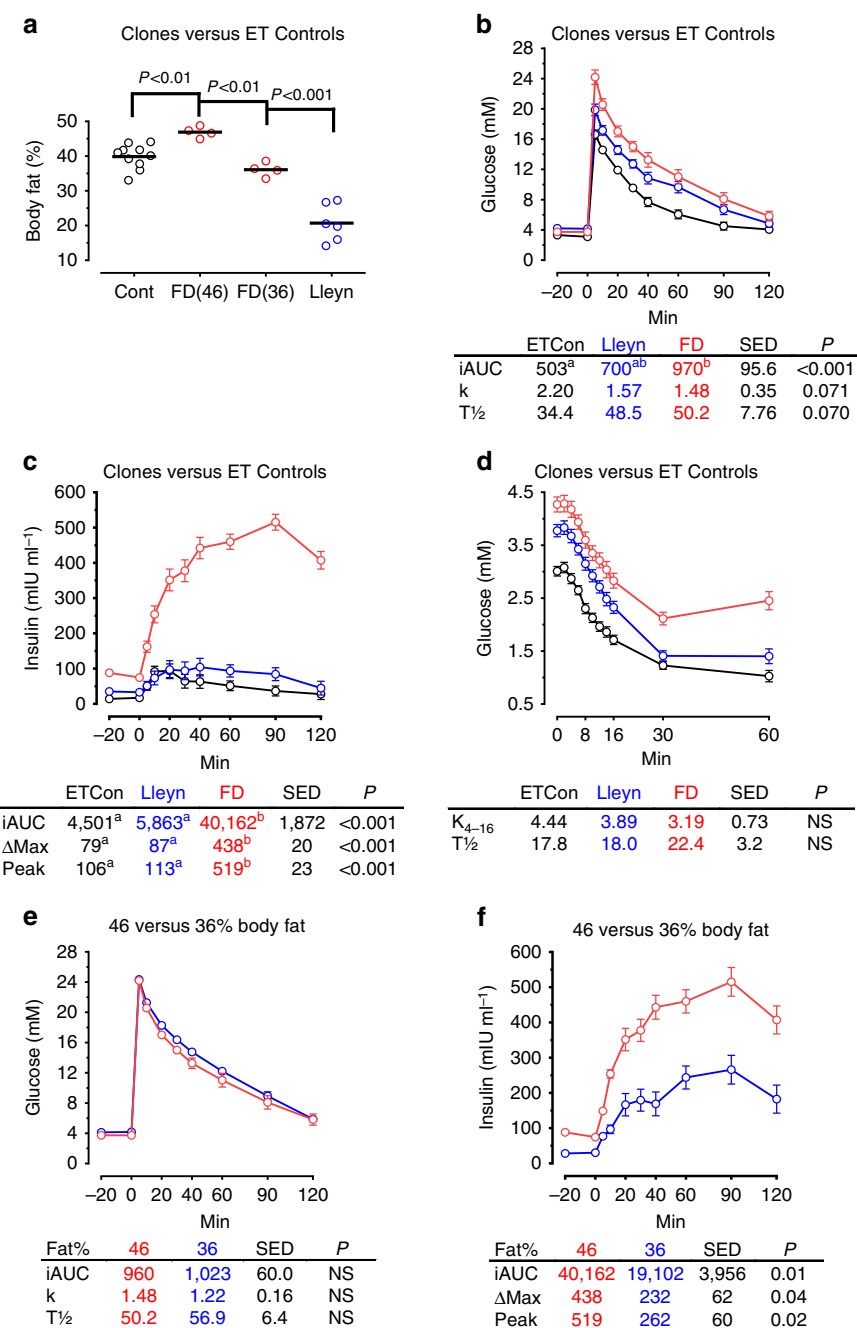

**Figure 2 | Body fat, glucose tolerance and insulin sensitivity in aged cloned sheep and contemporary embryo transfer (ET) controls.** Percentage body fat (determined by DEXA) revealed differences between clonal groups ($n=11$ ET-Controls, 6 Lleyn and 4 FD clones) (**a**). Finn-Dorset (FD) clones were subsequently placed on a diet to reduce body fat percentage from 46% (FD46) to 36% (FD36). Initial glucose tolerance tests revealed modest differences in glucose tolerance (**b**) and large differences in insulinaemia (**c**) between clonal groups that probably reflected genotype differences in body fat percentage and body fat distribution. Insulin sensitivity did not differ between clonal groups and ET-controls (**d**). Although glucose tolerance was unaltered (**e**), insulinaemia was significantly reduced when Finn-Dorset clones lost body fat (**f**). Glucose was infused i.v. at $0.4\,\mathrm{g\cdot kg^{-1}}$ body weight and insulin at $0.75\,\mathrm{IU\,kg^{-1}}$ body weight, both at time 0. iAUC, incremental area under curve; k, fractional clearance of glucose (% min$^{-1}$); $K_{4-16}$, % decline in glucose min$^{-1}$ from 4 to 16 min; T½, half-life of glucose; $\Delta_{Max}$, peak insulin—basal insulin. s.e.d., standard error of difference; NS, not significant. Different superscripts denote significance at $P<0.05$ following Bonferroni correction.

between genotypes[36]. Corresponding levels of body fat for clones 2264, 2265 (both female) and A089 (male) were 42%, 38% and 23%, respectively (Supplementary Table 2).

Non-fasted plasma glucose concentrations at the onset of the initial glucose tolerance test (GTT), 4 h after the morning meal, were all within the normal fasting range for well-fed sheep of moderate body condition (3.25 to 4.50 mmol/L (ref. 37)). The lower (ANOVA; $P < 0.001$; SED $= 0.24$) glucose concentrations (mmol/L) found in ET-Controls (3.20) than in either Finn-Dorset (3.70) or Lleyn (4.20) clones (Fig. 2b) are therefore of no clinical significance. The Finn-Dorset and Lleyn clones, nevertheless, appeared less tolerant to glucose infusion (400 mg kg$^{-1}$) than the ET-Controls, as evident from the incremental area under the curve (Fig. 2b). Resting plasma insulin ($\mu$IU ml$^{-1}$) was also greater (ANOVA; $P < 0.001$) for the Finn-Dorset clones (81.5) than for either the Lleyn clones (34.1) or ET-controls (15.7) at the onset of the initial GTT, and a hyperinsulinaemic state was confirmed in these animals following glucose infusion (Fig. 2c). Compared with fasted basal plasma insulin concentrations in lean and obese sheep from previous studies (Supplementary Table 3), these values for the Finn-Dorset clones appear high. However, neither percentage decline in plasma glucose per minute following insulin administration (0.75 IU kg$^{-1}$) ($K_{4-16}$), nor the half-time for glucose clearance t(1/2), differed between these animal groups (Fig. 2d), indicating that peripheral tissue insulin sensitivity was similar. To get a better sense of the contribution of body fat to insulin sensitivity, the four Finn-Dorset ewes were subsequently placed on an energy-restricted diet (0.7 × maintenance (ref. 33)) for 20 weeks during which time their level of body fat declined from ~46 to ~36% (Fig. 2a). Although glucose tolerance was not altered (Fig. 2e), both resting (29 $\mu$IU ml$^{-1}$) and glucose-stimulated plasma insulin concentrations were markedly reduced (ANOVA; $P < 0.05$) in the leaner Finn-Dorset clones (Fig. 2f), indicating that a major factor contributing to hyperinsulinaemia in these animals was level of body fat.

Corresponding non-fasting plasma glucose concentrations for the other three clones were close to or slightly above the normal fasting range (Supplementary Table 2). At ~38% body fat, the 8-year-old clone 2265 was fatter than her 7-year-old fellow Lleyn clones, but otherwise her indices of glucose and insulin tolerance were similar. The 9-year-old male clone was also euglycaemic, with glucose indices similar to that of ET-control males during the GTT. Although glucose tolerance was normal for this animal, insulin secretion ($\Delta_{Max}$) was low and peripheral tissue insulin sensitivity (that is, $K_{4-16}$ and half-life) was reduced relative to ET-Control males of similar body composition. Parameters for this animal, however, do not indicate a diabetic state.

**Ambulatory and stimulated blood pressure.** After recovery from telemeter implantation, resting blood pressure and heart rate were monitored over the next 4 days. No sheep exhibited any clear circadian rhythm of pressure (for example, nocturnal dipping analysed by non-linear regression using Fourier curves). Hence, Fig. 3 reports a composite pressure trace recorded over 4 days in which sheep were group-housed and exposed only to routine husbandry procedures. Mean arterial (MAP) and diastolic (DBP) blood pressure over the study period were similar among clones (MAP, 111 ± 7 versus 102 ± 8 mm Hg, $P = 0.18$; DBP, 96 ± 6 versus 90 ± 7 mm Hg, ANOVA; $P = 0.24$; Finn-Dorset versus Lleyns, respectively) and when compared with ET-controls (MAP, 114 ± 6: DBP, 106 ± 7 mm Hg). Systolic blood pressure was similar between clones and to ET-controls (Fig. 3a,b), but resting heart rate in ET-controls was lower (average HR, 76 ± 8 versus 91 ± 11 (Finn-Dorset) and 92 ± 6 (Lleyns) beats per min;

ANOVA; $P < 0.001$, Fig. 2c,d). Spikes in pressure and heart rate were noted in all sheep at morning feed time (~08:00 hours, Fig. 3a,c). Cardiovascular parameters recorded in the elderly clones were within the established normal range of pressures reported for aged sheep in the literature (Supplementary Table 4). The pressor response to incremental Angiotensin II (Ang II) was chosen as a cardiovascular challenge, as comparison with data from ET-controls and other studies conducted by us in sheep[38] were available for reference. Also, Ang II is a potent vasoconstrictor, often targeted as a cardiovascular therapy through the use of angiotensin-converting enzyme inhibitors. Step-wise (every 10 min) doubling of intravenous (i.v.) Ang II delivery incrementally increased BP in all sheep (Fig. 3e,f), the effect size being identical between clones (Finn-Dorset, 62 ± 12 versus Lleyn, 62 ± 12 mm Hg) and similar to ET-controls (58 ± 10 mm Hg). Heart rate tended to decrease with increasing Ang II infusion in Lleyn clones, but was refractory to the depressor effect of high blood pressure in Finn-Dorset clones (Fig. 3g), which was in keeping with the ET-controls (Fig. 3h).

**Musculoskeletal assessments.** Only two cloned sheep were identified as having an abnormal gait on clinical examination, scoring 1 and 2 on the six-point scale of Kaler et al[39]. (Supplementary Table 5). However no animal showed obvious signs of pain on manipulation of any joint. Radiographic assessment revealed that most sheep had either normal hip joints or mild grade 1 osteophytosis (Fig. 4). The only sheep with significant degenerative joint disease was the Finn-Dorset clone 2262, with moderate to severe osteophytosis (scoring 2–3 on the modified scale of Kellgren and Lawrence[40]) of the majority of joints (Figs 5 and 6a). MRI of the stifles demonstrated the degenerative changes apparent in the radiographs, with periarticular osteophytosis primarily on the trochlear ridges and caudal tibia (Fig. 6b,c). The severity of the changes was greatest in clone 2262. In addition, clones 2262 and 2263 had evidence of mild bilateral synovitis, while clones 2260 and 2261 had mild bilateral synovial effusions.

**Discussion**

This study represents the first detailed and comprehensive assessment of age-related, non-communicable diseases in adult cloned offspring of large animals. Working with sheep, four of which were cloned from the cell line that gave rise to Dolly, we undertook assessments of metabolic, cardiovascular and musculoskeletal health. Importantly, despite their advanced age (7–9 years), none of the clones showed any clinical signs of disease, being euglycaemic, insulin sensitive and normotensive. No animal was lame despite most showing radiographic evidence of mild OA in one or two joints, as would be expected in sheep of that age. None have required treatment for OA. In the absence of perfectly matched (by age, genotype and environment), naturally conceived controls, these conclusions were reached with reference to (a) a contemporary group of six-year-old sheep that were managed alongside the clones, and underwent the same metabolic and cardiovascular assessments, (b) published reference ranges for blood glucose, (c) pelvic radiographs of eight 5-year-old healthy sheep, and (d) the available scientific literature. Whilst contemporaneous controls would have been ideal, our data are nevertheless compelling, indicating no detrimental long-term adverse effects of SCNT on the health of aged adult offspring.

Because cloning by SCNT can lead to obesity and hyperinsu-linaemia in aged mice[18], and a single case of Type I (insulin dependent) diabetes was reported in a cloned calf[41], we decided to measure glucose tolerance and insulin sensitivity in our aged cloned offspring. There are (infrequent) reports of naturally

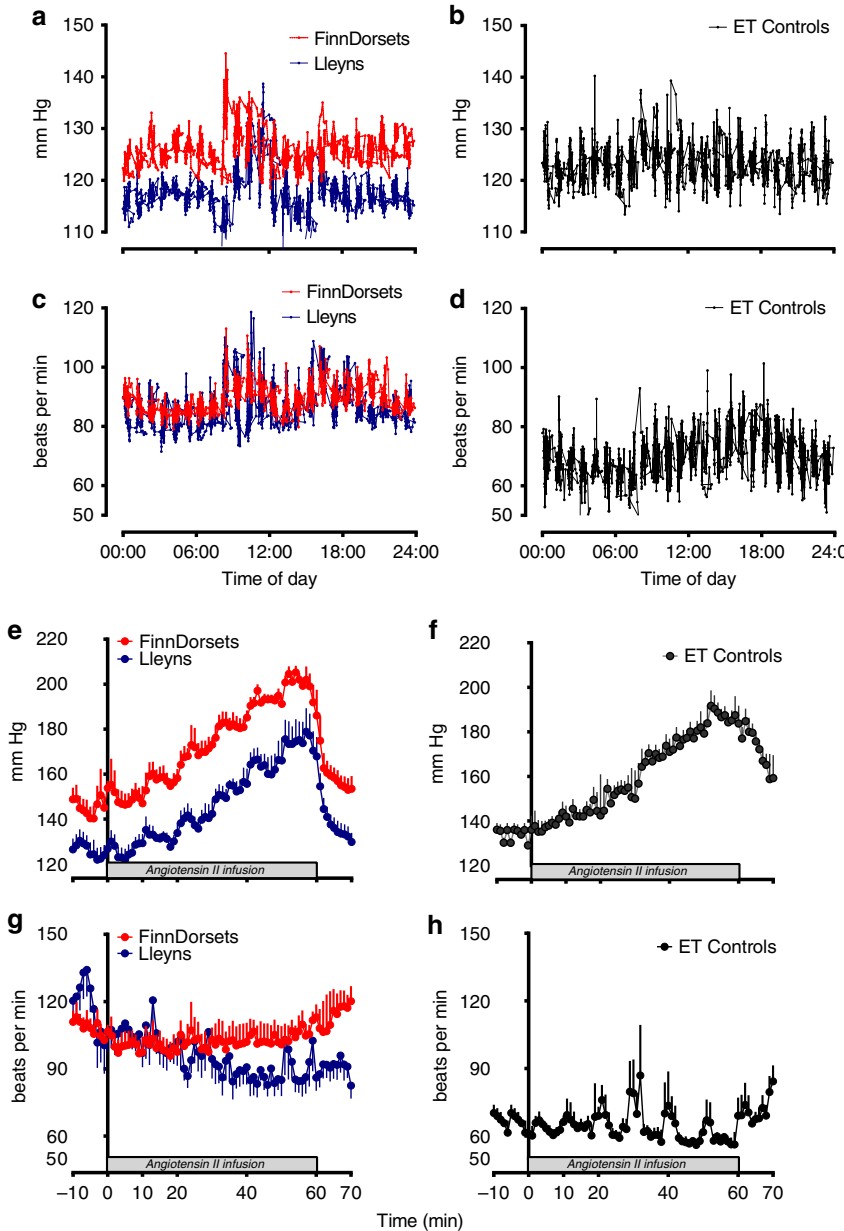

**Figure 3 | Ambulatory and Angiotensin-II stimulated systolic blood pressure and heart rate in Finn-Dorset (n = 4) and Lleyn (n = 6) clones, and embryo transfer controls (ET-controls; n = 11).** Cardiovascular data were recorded using signals derived from radiotelemetry probes (Telemetry Research, distributed by Millar Ltd, Oxon, UK) implanted into the carotid artery. Data are presented as minute means ± s.e.m. and represent either composite 24 h traces of ambulatory systolic blood pressure and heart rate recorded over 3–4 days with sheep group-housed in a floor pen (**a,b,c,d**) or the cardiovascular response to incremental i.v. angiotensin II infusion (0–60 ng kg$^{-1}$ min$^{-1}$) (**e,f,g,h**). Wireless sampling was at 2 kHz (that is, beat-to-beat), data downloaded automatically to acquisition software on a laptop (LabChart 7; AD Instruments, Oxon, UK). Shaded box represents period of angiotensin II infusion.

occurring type I diabetes in sheep[42], where glucose concentrations for affected animals range between 16 to 19 mmol l$^{-1}$. However, we are unaware of cases of type II (non-insulin dependent) diabetes, although insulin resistance has been extensively studied in this species[43] where it is known, for example, that deficiencies in maternal periconceptional diet can lead to insulin resistance in adult offspring[44]. In humans, risk factors for insulin resistance leading to type II diabetes include advanced age and obesity; the latter risk factor is also related to enhanced insulin resistance in sheep[43,45]. Hence the current study sought to create an obese state in our experimental animals to fully test the extent of glucose tolerance and insulin sensitivity[46]. Hyperinsulinaemia in the four Finn-Dorset clones (Fig. 2c) was

certainly influenced by the level of body fat, as weight loss reduced plasma insulin concentrations (Fig. 2f), consistent with improved peripheral insulin sensitivity which can be expected following weight loss in sheep[46]. However, peripheral insulin sensitivity is negatively related to visceral fat and intrahepatic lipid accumulation in humans[47,48], which varies between ethnic groups[49]. It follows that genotype differences in regional fat distribution in sheep[36] may at least partially account for the large difference in insulin response following glucose infusion observed between Finn-Dorset clones and ET-controls, which had a similar proportion of total body fat (Fig. 2a). Although this awaits final confirmation following post-mortem examination, previous studies indicate that the Finn-Dorset genotype has larger

intra-abdominal fat depots than other common genotypes such as our ET-controls[50,51].

Consistent with observations for cloned offspring from other species[9,14,52], perinatal losses of cloned sheep have been associated with structural defects of the kidneys and heart, and with pulmonary hypertension[10]. Enlarged and abnormally developed kidneys were also evident in some of the six live-born Finn-Dorset clones that failed to survive to 3 months of age in the current study (Supplementary Table 6). We hypothesized that if subtle renal defects in surviving offspring were to persist late into adult life then this could lead to increased blood pressure, known as reno-vascular hypertension[53]. Hypertension has been described to 'follow-the-kidney' in cross-transplantation

studies, suggesting factors intrinsic to the kidney mediate increases in blood pressure[54–56]. The renin–angiotensin–aldosterone system is a major target for long-term anti-hypertensive treatment as elevated intra-renal renin–angiotensin–aldosterone system action and/or sensitivity may directly cause increased blood pressure[57,58]. Here, for the first time using radiotelemetry blood pressure probes, we recorded circadian cardiovascular parameters continuously for individual cloned and ET-control sheep kept as a group in a stress-free environment. Ambulatory blood pressure of these cloned animals was similar to ET-controls and within the normal range for other sheep reported in the literature (that is, ~120/80 mmHg; Supplementary Table 4). It was also similar to adult blood pressure in healthy humans. Classification of 'hypertension' in humans begins at a systolic blood pressure ≥140 mmHg (ref. 59) or ~20 mmHg above resting. The elderly cloned sheep in this study could not, therefore, be considered to be hypertensive. Direct i.v. infusion of angiotensin II, to further challenge cardiovascular responsivity, failed to induce a greater pressor response than in ET-controls. We therefore conclude that cloning by SCNT had no detrimental, latent effect on the cardiovascular system of sheep in this study that survived into late adulthood.

Detailed musculoskeletal investigations were undertaken of all sheep, as Dolly had required treatment for OA from around 5 years of age, raising concerns over premature ageing in cloned animals[22–24]. Radiographs taken of Dolly showed OA in the left stifle (knee), however OA was evident in both stifles at post-mortem examination 18 months later (TJ King, Roslin Institute, personal communication). The consensus at that time was that, because the OA was localized rather than generalized, it was more likely of traumatic aetiology. However, the aetiopathogenesis of OA is accepted to be multifactorial, involving both genetic and acquired factors (such as joint trauma/overloading, and obesity). The prevalence increases with age (the strongest risk factor), but the course of the disease varies greatly between individuals[60,61].

Clinical examination of cloned sheep in the present study revealed only mild lameness in the left foreleg of one animal, which may have been associated with her slightly abnormal forelimb conformation. None of the Finn-Dorset clones had an abnormal gait or showed lameness. Several clones had soft tissue

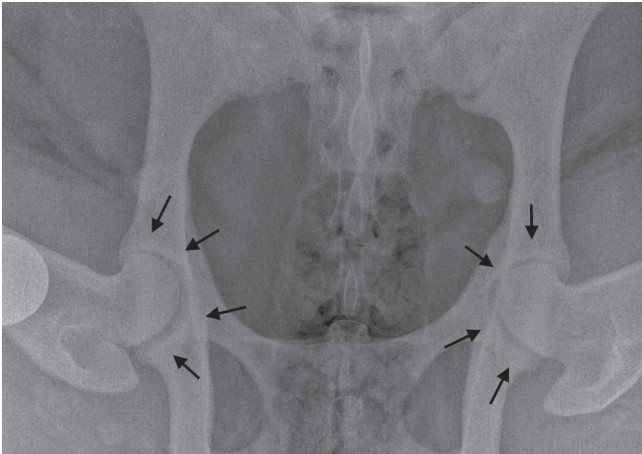

**Figure 4 | Image showing a ventrodorsal radiographic view of the hip joints of clone 2260.** Both femoral heads appear normally seated within the acetabulae, with minimal remodelling and no obvious osteophytosis. A mild increase in bone density (white) is evident around both acetabular rims (black arrows), however there is no significant remodelling of the acetabulum or femoral head and neck regions. This sheep therefore has minimal radiographic evidence of osteoarthritis. The circular opacity is a 25 mm coin, placed as marker to indicate the right limb.

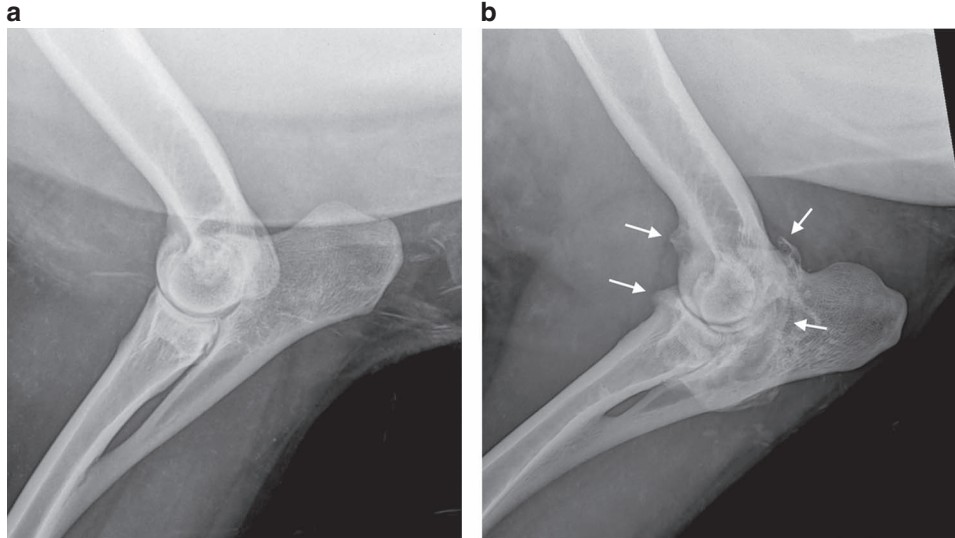

**Figure 5 | Mediolateral radiographic views of the elbow joint of two clones with and without osteoarthritis. (a)** The elbow joint of Clone 388 appears normal: there is no radiographic evidence of osteoarthritis. **(b)** The elbow joint of Clone 2262 shows evidence of severe osteoarthritis. Large osteophytes representing new bone formation are apparent at the joint margins (white arrows), and are associated with bone remodelling. This elbow was scored as Grade 3 on the modified scale of Kellgren and Lawrence[40].

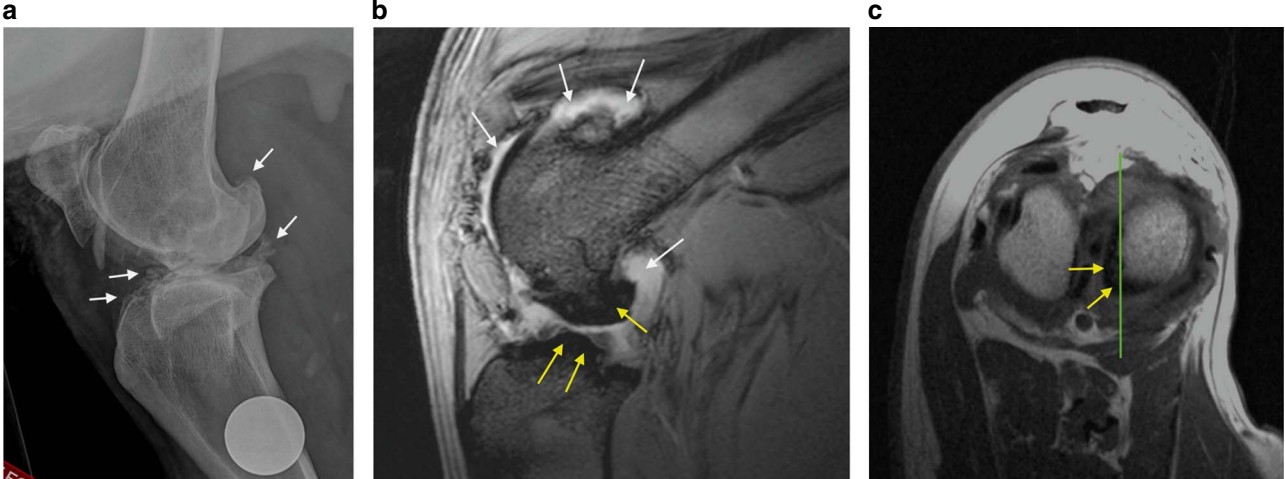

**Figure 6 | Images of the right stifle joint of Clone 2262.** (**a**) Mediolateral radiographic view of the right stifle of Clone 2262. White arrows indicate osteophytes/new bone formation at joint margins, with bone remodelling. This joint was scored as Grade 3 on the modified scale of Kellgren and Lawrence[38]. The circular opacity is a 25 mm coin, placed as marker to indicate the right limb. (**b**) Sagittal T2* MRI scan of the same stifle joint. (**c**) Transverse T1 MRI scan of the same stifle joint. The green lines on this image show the level corresponding to the sagittal image shown in **b**. In the MRI scans, bone appears hypointense (dark), while fat and fluid appear hyperintense (white). The irregular new bone formation on the femur and tibia are indicated by the yellow arrows. The white arrows indicate synovial fluid and fat within joint.

swelling or fibrous thickenings of various parts of their limbs, as would be expected in a random sample of sheep. However, these did not appear to be associated with any discomfort. Subsequent radiography revealed mild OA in various joints in several sheep. It is not surprising that no sheep showed marked lameness, as radiographic changes indicative of OA do not necessarily correlate with the extent of clinical disease[62]. Mild OA (scored by two independent reviewers) was seen most often in the hip (13 of 13 sheep) and stifle (12 of 13 sheep) joints. Scott[63] reported that the stifle and elbow joints are most commonly affected by joint trauma. However in the present study, only 5 of 13 sheep showed elbow OA, which was bilateral in most cases. Only Clone 2262 had significant OA in multiple, but not all, joints. The majority of sheep appeared to have a mild increase in bone density of the acetabular rim, however none had significant osteophytosis or remodelling of the acetabulum or femoral head and neck as would be expected with OA. Similar findings were apparent in pelvic radiographs of eight healthy 5-year-old sheep (Supplementary Fig. 2), enroled in an unrelated study, made available courtesy of Professors Allen Goodship and Gordon Blunn (University College London). Thus it may be a normal radiographic finding in healthy ageing sheep, and not, in isolation, indicative of OA. This will be further assessed when post-mortem examination of the present cohort is undertaken at a future time.

Radiographic changes appear relatively late in the disease process and so MRI of the stifles (knees) of the four Finn-Dorset clones was undertaken, as MRI is more sensitive in detecting early structural changes[60]. To minimize the duration of sedation required, MRI scans were taken only of the stifle joints, because these were among the most commonly affected joints in our population, and were affected in Dolly[23]. T2* and proton density, fat-saturated MRI images provided the most useful diagnostic information, showing periarticular osteophytosis primarily on the trochlear ridges and caudal tibia. However, the extent of the disease remained mild, and not unexpected in aged sheep.

Reports of OA in non-experimental sheep are rare in the literature. A case series was reported by Scott[64], describing elbow OA in 10 sheep (aged 2 to 5 years) that had shown forelimb lameness of at least 3 months duration. A post-mortem study of

the stifle joints of 65 clinically normal sheep aged 6 months to 11 years by Vandeweerd et al.[65] identified cartilage defects in 66% of the sheep, the severity of the lesions increasing with age. These authors proposed that OA exists in normal ageing sheep, and is not always associated with clinical signs. Pathological changes were also found in the femoral head cartilage of clinically normal 2.5- to 3-year-old sheep by Zilkens et al[66]. Thus there is no evidence of an increased incidence or severity of OA in the clones in this study. However, a single cloned sheep in the present study had obvious OA in multiple joints, although it varied in severity between joints, and not all joints were affected. It is not surprising that the joints were variably affected, as studies in humans have shown, for example, that a factor such as obesity increases the risk of knee OA by a factor of three, but has less effect on hip OA[60]. As discussed previously, a more homogenous distribution might have been expected, both within and between sheep, if cloning by SCNT was a direct causative factor in the development of OA.

From the current series of assessments we conclude that there are no long-term detrimental health effects of cloning by SCNT for a long-lived species such as the sheep. This conclusion is consistent with less detailed longevity studies in cattle[17], and suggests that the ageing process in surviving clones of large animal species is not accelerated. On initial inspection, our data in sheep may appear at odds with the health status of Dolly and predictions of premature ageing, which were based on terminal restriction fragment analyses of her genomic DNA[20]. While telomere length was reduced in SCNT clones relative to age-matched controls in that and subsequent studies in sheep[67], these effects did not manifest following SCNT in cattle[21]. Further inconsistent reports of shorter telomeres in cloned offspring from other species[68] have led to the consensus that telomere length is generally restored during nuclear reprogramming[69]. The extent of telomere restoration in turn is dictated by intricate epigenetic alterations to telomeric and sub-telomeric chromatin, variation in which could explain discrepancies between species and donor cell types within species. It follows that the relationship between telomere length, health and longevity in multicellular organisms is complex and, for our current cohort of animals, awaits organ-specific cell enrichment and analysis following post-mortem at a future date.

In conclusion, although the efficiency of SCNT has improved in recent years, its overall efficiency remains low, with high embryonic and gestational losses compared to natural mating and assisted reproduction. A relatively high proportion of clones also fail to successfully make the transition to extra-uterine life, some harbouring congenital defects, such as observed in the kidney. For those clones that survive beyond the perinatal period, however, the emerging consensus, supported by the current data, is that they are healthy and seem to age normally.

## Methods

**Animal work.** Animal work was conducted under the authority of the United Kingdom Animal (Scientific Procedures) Act 1986, and approval was obtained from the University of Nottingham Animal Welfare and Ethical Review Board before commencement of these studies.

**Genetic identity of cloned sheep.** This was confirmed using 21 microsatellite markers including those recommended by the International Society of Animal Genetics (Supplementary Table 1). Sheep blood samples and cell lines were submitted for PCR and fragment analysis to an independent genetic Company (VHL Genetics, The Netherlands). PCR amplification and fragment analysis was conducted in an ABI PRISM 3130XL automatic sequencer (Applied Biosystems), and the amplified fragments were classified using the GENESCAN 3.7 software (Applied Biosystems). We undertook a blinded pairing of cloned sheep and parental cells, and all clones matched their cells of origin for all 21 markers (CERVUS v.3.0 software[70]).

**Reference population (ET-controls).** This consisted of a group of 6-year-old virgin male ($n = 6$) and female ($n = 11$) Suffolk x Scottish Blackface sheep generated as part of the study of Sinclair et al.[44]. These animals were the offspring of the control group of normally fed ET donors in that study, and were accommodated in pens adjacent to the cloned offspring where they were fed the same diet. Health assessments were originally reported for these animals at two years of age[44]. Further assessments of body composition, glucose and insulin tolerance, and blood pressure were undertaken again for these animals at 6 years of age using protocols described herein.

**Whole animal body composition.** Glucose tolerance and insulin sensitivity are each challenged in the obese state[43], which is why we endeavoured to create this condition in both our reference population of ET-controls and clones. Animals were fed a proprietary concentrate (Manor Farm Feeds, Owston, Leicestershire, UK) consisting 60 g kg$^{-1}$ oil, 145 g kg$^{-1}$ crude protein, 80 g kg$^{-1}$ fibre and 65 g kg$^{-1}$ ash, in a mixed diet with meadow hay (dry matter (DM), 830 g kg$^{-1}$; metabolizable energy, 8.5 MJ kg$^{-1}$ DM; crude protein, 95 g kg$^{-1}$ DM). This diet was fed on an ad libitum basis. To standardize body composition, body condition scores (BCS) ($1_{[thin]}$ to $5_{[obese]}$ scale[34]), were obtained at fortnightly intervals for these animals with the aim of attaining a BCS of 4.0 units at the time of metabolic assessments, equivalent to approximately 40% total body fat in mature female ewes[35]. Formal assessments of body composition were assessed by DEXA using a validated in-house protocol[71]. Animals were sedated (intramuscular (i.m.) 0.1 mg kg$^{-1}$ Xylazine; i.v. 4 mg kg$^{-1}$ Ketamine) and scanned transversally along the longitudinal axis using a Norland XR-800 DXA scanner. The scan lasted ~15 min after which animals were returned to the barn.

**Glucose and insulin tolerance tests.** Glucose tolerance and insulin sensitivity were determined using standard in-house protocols[44,71]. Briefly, for the GTT, basal plasma insulin and glucose concentrations were established 4 h after the animals' morning meal. Infusion of sterile glucose (i.v. 0.4 g · kg$^{-1}$ body weight given as a single bolus) was followed by blood sample collection from indwelling jugular catheters into K$^+$ EDTA vacutainers at 5, 10, 20, 30 40, 60, 90 and 120 min. The insulin tolerance tests were carried out before the morning meal. Infusion of insulin (0.75 IU kg$^{-1}$) was followed by blood collection into K + EDTA vacutainers at two-minute intervals up to 16 min, and then again at 30 and 60 min. Plasma glucose concentrations were determined on a Cobas Miras Plus autoanalyzer (ABX Diagnostics); the kit for glucose was supplied by Randox Laboratories (catalogue no. GL2623). Plasma insulin concentrations were determined using an ovine specific insulin ELISA kit (Mercodia AB, Uppsala, Sweden).

**Radiotelemetric assessments of heart rate and blood pressure.** Sheep were anaesthetised for general surgery using i.v. ketamine (2.0 mg kg$^{-1}$) and midazolam (0.3 mg kg$^{-1}$), dosed to effect for intubation. Anaesthesia was maintained with isoflurane (1–2% in O$_2$). Solid-state pressure radiotelemeters designed for large animal use (TRM84PB; Telemetry Research, NZ and distributed in the UK by Millar Inc, Oxon, UK) were inserted (10 cm) into the carotid artery with the body of the transmitter secured subcutaneously in the neck (Supplementary Fig. 1).

The incision site was closed and the animal recovered to a pen. Post-operative antibiotics were administered (i.m. long-acting amoxycillin) and analgesia maintained with a fentanyl (0.2 mg kg$^{-1}$) patch fixed to the forelimb. Sheep (4 on each occasion) were recovered to a floor pen and wireless recording of individual cardiovascular parameters (for example, blood pressures, heart rate) began immediately for 3–4 days for each sheep. Continuous recording at a sampling rate of 2 kHz was pre-programmed for 10 min of each hour, all day and night for all four sheep in each batch using an automated scheduler pod linked to data acquisition software (LabChart 7; AD Instruments, Oxon, UK). On the final day, all sheep were challenged with a step-wise incremental Angiotensin II (0–60 ng kg$^{-1}$ min$^{-1}$; Sigma-Aldrich, UK) infusion, delivered into the jugular vein after earlier venipuncture and restraint in a metabolic crate. Recording at this time was of individual sheep on a second-by-second basis. Sampled data were stored, analysed and minute means derived (after exclusion of artefacts) using LabChart 7 and MS Excel. Telemeters were surgically removed from all sheep under general anaesthesia the following day, following similar protocols to described above. The solid-state telemeters were used (after recalibration and sterilization) on three separate occasions, to obtain cardiovascular data for ten sheep ($n = 4$, Finn-Dorset clones; $n = 6$, Lleyn clones). Cardiovascular protocols were successfully completed in all sheep.

**Clinical examination and diagnostic imaging of joints.** All sheep underwent a full musculoskeletal examination by a veterinary orthopaedic specialist (co-author SC) involving visual assessment of gait, followed by palpation and manipulation of the limbs and spine. Palpation assessed for evidence of muscle atrophy, swelling, heat or pain in the limb. Manipulation assessed individual joints for range of motion, pain or crepitus. The feet were also examined for pathology. Assessment of gait was made using the seven-point scale (0 to 6) developed by Kaler et al[39].

All sheep were subsequently sedated (i.m. 0.1 mg kg$^{-1}$ Xylazine; i.v. 4 mg kg$^{-1}$ Ketamine) for 10 to 15 minutes to obtain mediolateral radiographs of both hocks, stifles, elbows and carpi, and a ventrodorsal view of the pelvis, including both hips and the lumbosacral junction. Shoulders were radiographed but the images found to be of inconsistent quality for valid interpretation. Radiographs were taken using a portable Tru-DR Digital Radiography System, with laptop (R8-51D45) and plate (50-C907-08B) (BCF Technology Ltd, Strathclyde, UK), and analysed using OsiriX Lite (http://www.osirix-viewer.com/download_form/download_form.php). Images were assessed for evidence of OA, and graded using a modified Kellgren and Lawrence[40] scale. Assessment of joint space width was not undertaken as the sheep were sedated and recumbent (that is, non-weight-bearing) when the radiographs were taken. Radiographs were scored independently by two veterinary orthopaedic specialists (co-author SAC, and Malcolm Ness, independent) who were unaware of sheep identity. OA was graded based on the presence of osteophytes and bone remodelling, where 0 = no evidence, 1 = questionable, mild osteophytosis, 2 = moderate osteophytosis and minor bone remodelling, and 3 = severe osteophytosis with definite bone remodelling. The four Finn-Dorset clones underwent MRI of their stifle joints, using a mobile MRI scanner (Burgess Diagnostics Ltd. Earnshaw Business Centre, Hugh Lane Leyland PR26 6PD), with a high-field 1.5Tesla magnet (Philips Medical Systems, Achieva). Sheep were induced using i.v. ketamine (2.0 mg kg$^{-1}$) and midazolam (0.3 mg kg$^{-1}$), dosed to effect for intubation. Anaesthesia was maintained with isoflurane (1–2% in O$_2$). Scans were obtained using a SENSE Flex-S receiver coil and Body Transmit Coil, with a 130mm Field of View. Transverse T1, Saggital T2*, PD and PDFS protocols used to obtain images extending from 2 cm proximal to the patella, to 5 cm distal to the tibial plateau. Images were transferred in DICOM format on CD and subsequently analysed using OsiriX Lite.

**Statistical analyses.** Because the six Lleyn clones were derived from a study that assessed the effects of Xenopus oocyte extract before nuclear transfer on pregnancy outcomes[30], we initially investigated if this had any influence on our measurements of body composition, glucose metabolism and blood pressure by ANOVA. As the extract had no effect we treated these animals as a single cohort in all subsequent analyses. Glucose and insulin tolerance tests were subsequently analysed by repeated-measures ANOVA (with Bonferroni correction for comparisons between means) where clone identity (Finn-Dorset, Lleyn) and time formed the fixed effects. Single values for incremental areas under the response curves were generated for each individual using the trapezoid rule in Prism 6 (GraphPad Software, La Jolla, CA, USA) and were analysed by one-way ANOVA with Bonferroni correction for comparisons between means. For cardiovascular analyses, data were either analysed as summary measures (e.g. minute means of sec-sec radiotelemetry pressures or heart rate) and analysed similarly to metabolic data or, to test for any circadian effects, by incorporating all recorded cardiovascular datapoints (~2,500–4,500 for each animal) into a non-linear regression model fitting a Fourier-curve ($Y = \alpha + \text{ß}sin(2\pi(X + \varepsilon)/w)$ to derive four parameters $\alpha$, set-point; $\beta$, amplitude; $w$, wavelength and $\varepsilon$, offset. No sheep exhibited any circadian cardiovascular rhythm (pressure or heart rate) and therefore only baseline (average for each variable) or minute means of responses (for example, to angiotensin II infusion) are presented. All data were analysed using Genstat v18 (VSNi, Rothampsted, UK) and are presented as means ± s.e.m. or s.e.d. (standard error of the differences between means, for a more conservative estimate of the contrast variance).

**Data availability.** The data that support the findings of this study are available from the corresponding author (KDS) on reasonable request.

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

## Acknowledgements

This article is dedicated to the memory of co-author Keith H.S. Campbell (1954 to 2012). The University of Nottingham provided financial support for this study. Professors Gordon Blunn (University College London) and Allen Goodship (University College London and Royal Veterinary College) for making available pelvic radiographs of 5-year-old healthy sheep. Malcolm Ness, DipECVS, FRCVS (Longframlington, Northumberland, UK) for independent assessment of radiographs. Professor CG Gutierrez received financial support from DGAPA, UNAM, towards a sabbatical visit to the University of Nottingham.

## Author contributions

K.D.S. conceived and designed the health-assessment experiments. K.H.C., P.A.F., J.-H.L., A.J.R. and I.C. cultured donor cells, generated cloned offspring and provided details of pregnancy outcomes. K.D.S., D.S.G., S.A.C. and C.G.G. undertook metabolic, cardiovascular and musculoskeletal assessments, and analysed the data. K.D.S., D.S.G., S.A.C. and C.G.G. wrote the manuscript.

## Additional information

**Competing financial interests**: The authors declare no competing financial interests.

**How to cite this article**: Sinclair, K. D. *et al.* Healthy ageing of cloned sheep. *Nat. Commun.* 7:12359 doi: 10.1038/ncomms12359 (2016).

