## [Peer Review file · Nature Communications]

Reviewers' Comments:

Reviewer #1 (Remarks to the Author)

The manuscript NCOMMS-16-08113 aims to assess the late-onset non-communicable diseases in cloned sheep. The subject is relevant and the data presented are original.

Involved methodology is adequate and presented data are properly elaborated and are valid. Very convincing images give additional value to the manuscript.

The statistical methods are appropriate.

Presented data represents a comprehensive assessment of age-related diseases in clones. The discussion is adequately structured, relevant references are used and the conclusions are valid.

Reviewer #2 (Remarks to the Author)

Review of manuscript NCOMMS-16-08113

Title: "Healthy aging of cloned sheep"

Authors: Sinclair et al.

Abstract

The present study investigated several physiological parameters related to aging in sheep derived either by somatic cloning or by conventional embryo transfer. The authors did not find any differences with regard to glucose tolerance, insulin sensitivity, blood pressure, incidence of osteoarthritis, and other degenerative joint diseases between the two groups of sheep. The authors conclude that these data indicate that SCNT does not have long-term detrimental effects on clones.

Criticism

From the birth of the first cloned animal, "Dolly" the sheep, questions related to putative long-term effects and longevity of cloned animals have been raised as a concern to use SCNT in animal production. A number of studies have convincingly shown that clones and their offspring are normal compared to conventionally produced animals. But elevated embryonic and fetal losses have been reported and also higher losses in the perinatal phase. Some studies in the laboratory mouse have shown that cloned mice may show obesity and a pathological phenotype, and in addition also had a shorter lifespan. The present paper reports the results of a comprehensive study on sheep, specifically looking at putative long-term effects of the somatic cloning process. Such results in large animals are still lacking. The current study used aged cloned sheep (7 to 9 years old) that were subjected to an extended clinical investigation, including glucose tolerance, insulin sensitivity, blood pressure, and analysis of degenerative joint diseases. Data from the cloned animals were compared to data from 5 to 6 years old sheep that had been produced by ET. The following weaknesses apply:

1. The main weakness of this study is that the authors do not have age and breed matched controls, but instead have used animals that were younger and from a different breed. Despite this limitation, the current data convincingly show that aged cloned sheep show normal physiology as compared to the control group. However, the authors address this limitation correctly in the first paragraph of Discussion.
2. The aging of animals produced by somatic cell nuclear transfer has been discussed previously, e.g. Exp. Gerontol. 2002, Kühholzer-Cabot and Brem. In addition, the authors should refer to the current knowledge on telomere biology in the context of aging and cloning as this is closely related to longevity.
3. The authors should somewhere in the paper mention the sex of the clones. Probably, they were all female, but it should be made clear in the Materials and Methods.
4. Any data from epigenetic analyses of selected genes/tissues would have been valuable in this context.

In conclusion, the present study reports valuable and novel information in the context of somatic cloning and aging, which is critical for the assessment of long-term effects of this technology. The paper is well written and the authors have used adequate methodologies and interpret their data with the necessary caution. After acceptance of the above points, I can see no reason to delay this publication.

Reviewer #3 (Remarks to the Author)

See attached file.

Reviewer #4 (Remarks to the Author)

This is a manuscript reports on a unique population, and the conclusion that the cloned animals are metabolically healthy and do not have OA is an important finding.

The manuscript reports on a novel animal population, and as such the findings are novel.

The manuscript is primarily descriptive, and as such could be misleading in its conclusion. Since a small number of animals underwent a large number of analyses, from a statistical standpoint drawing any conclusions are problematic. The study likely does not have the power to detect anything but the most drastic differences when so many parameters are analyzed. As such, the conclusions and results should be modified to also include what the power of this study would have to detect each parameter using a Bonferoni correction for multiple variables. While the use of a standard diet is useful in glucose tolerance analysis, it is not clear what the role in OA and bone health might be.

The detailed quantitative analysis of the radiographs and MRI analysis should be undertaken using measures taken by multiple individuals with radiographic analysis expertise blinded to the source of the animals. OARSI has developed standards for such measures that should be used, and these include cartilage volume for the MRI studies, and joint space analysis on the radiographs.

Reviewer #3:

Healthy ageing of cloned sheep

Sinclair et al.

In this manuscript the authors report the health status of 13 sheep generated by somatic cell nuclear transfer (SCNT) between the ages of 7 and 9 years old. Endpoints reported include glucose tolerance, insulin sensitivity, ambulatory and stimulated blood pressure, and radiological examinations of all joints.

When compared to non-cloned aged-matched controls, no significant differences in these parameters were observed, with the exception of aged clones Finn-Dorset that showed a decreased glucose tolerance, and one sporadic case of joint disease in a Finn-Dorset clone.

The authors have presented compelling evidence showing that sheep generated by SCNT, once surviving the perinatal period, can be as healthy as any animal produced by fertilization.

A minor criticism of this study is the fact that the authors do not mention the possibility of using molecular markers of aging such as telomere length in blood cells or the capacity of these animals to respond to immune challenges (cytokine measurements).

This reviewer can guess the reason why these experiments were not performed, i.e. the lack of reference data from non-cloned age-matched controls, however, the authors should mention why these - more molecular - endpoints were not used.

Finally, the authors should also state that the ultimate endpoints remain to be measured i.e. health-span and life-span.

Otherwise the manuscript deserves to be published without further experiments.

Response the Reviewers

Reviewer #1 (Expert in assisted reproduction of veterinary animals; Remarks to the Author):

Comment: The manuscript NCOMMS-16-08113 aims to assess the late-onset non-communicable diseases in cloned sheep. The subject is relevant and the data presented are original. Involved methodology is adequate and presented data are properly elaborated and are valid. Very convincing images give additional value to the manuscript. The statistical methods are appropriate. Presented data represents a comprehensive assessment of age-related diseases in clones. The discussion is adequately structured, relevant references are used and the conclusions are valid.

Response: We acknowledge the reviewers appreciation of the originality and value of the current study.

Reviewer #2 (Expert in livestock biotechnology; Remarks to the Author):

Abstract

The present study investigated several physiological parameters related to aging in sheep derived either by somatic cloning or by conventional embryo transfer. The authors did not find any differences with regard to glucose tolerance, insulin sensitivity, blood pressure, incidence of osteoarthritis, and other degenerative joint diseases between the two groups of sheep. The authors conclude that these data indicate that SCNT does not have long-term detrimental effects on clones.

Criticism

From the birth of the first cloned animal, "Dolly" the sheep, questions related to putative long-term effects and longevity of cloned animals have been raised as a concern to use SCNT in animal production. A number of studies have convincingly shown that clones and their offspring are normal compared to conventionally produced animals. But elevated embryonic and fetal losses have been reported and also higher losses in the perinatal phase. Some studies in the laboratory mouse have shown that cloned mice may show obesity and a pathological phenotype, and in addition also had a shorter lifespan. The present paper reports the results of a comprehensive study on sheep, specifically looking at putative long-term effects of the somatic cloning process. Such results in large animals are still lacking. The current study used aged cloned sheep (7 to 9 years old) that were subjected to an extended clinical investigation, including glucose tolerance, insulin sensitivity, blood pressure, and analysis of degenerative joint diseases. Data from the cloned animals were compared to data from 5 to 6 years old sheep that had been produced by ET.

The following weaknesses apply:

1. The main weakness of this study is that the authors do not have age and breed matched controls, but instead have used animals that were younger and from a different breed. Despite this limitation, the current data convincingly show that aged cloned sheep show normal physiology as compared to the control group. However, the authors address this limitation correctly in the first paragraph of Discussion.

Response: We agree with the reviewer and were aware of the limitations of this study from the outset. We therefore sought to develop an honest and compelling case based on use of available animal resources at our university and reference to relevant published literature.

2. The aging of animals produced by somatic cell nuclear transfer has been discussed previously, e.g. Exp. Gerontol. 2002, Kühholzer-Cabot and Brem. In addition, the authors should refer to the current knowledge on telomere biology in the context of aging and cloning as this is closely related to longevity.

Response: We have now added a section towards the end of the manuscript that briefly discusses telomere biology in the context of SCNT. We were aware of the review referred to above at the time of submission, but have since elected to cite two more recent reviews and an original research article in this new section.

3. The authors should somewhere in the paper mention the sex of the clones. Probably, they were all female, but it should be made clear in the Materials and Methods.

Response: In fact sex of the different clones is referred to in the main body of the manuscript and in Supplementary Tables 1 (microsatellite analyses) and 2 (Body composition and metabolism). We now make further reference to animal sex in the text of the manuscript (highlighted in red) and in Supplementary Table 5 (Musculoskeletal assessments).

4. Any data from epigenetic analyses of selected genes/tissues would have been valuable in this context.

Response: To be meaningful, such analyses would ideally require access to enriched populations of cells from several tissues, representing the different germ layers, which can only be attained in sufficient quantity post mortem. We have in mind to undertake such analyses at a future date following necropsy. At the time of writing, all 13 clones are alive and their health status is being monitored.

In conclusion, the present study reports valuable and novel information in the context of somatic cloning and aging, which is critical for the assessment of long-term effects of this technology. The paper is well written and the authors have used adequate methodologies and interpret their data with the necessary caution. After acceptance of the above points, I can see no reason to delay this publication.

Reviewer #3 (Expert in SCNT; Remarks to the Author):

In this manuscript the authors report the health status of 13 sheep generated by somatic cell nuclear transfer (SCNT) between the ages of 7 and 9 years old. Endpoints reported include glucose tolerance, insulin sensitivity, ambulatory and stimulated blood pressure, and radiological examinations of all joints. When compared to non-cloned aged-matched controls, no significant differences in these parameters were observed, with the exception of aged clones Finn-Dorset that showed a decreased glucose tolerance, and one sporadic case of joint disease in a Finn-Dorset clone. The authors have presented compelling evidence showing that sheep generated by SCNT, once surviving the perinatal period, can be as healthy as any animal produced by fertilization. A minor criticism of this study is the fact that the authors do not mention the possibility of using molecular markers of aging such as telomere length in blood cells or the capacity of these animals to respond to immune challenges (cytokine measurements). This reviewer can guess the reason why these experiments were not performed, i.e. the lack of reference data from non-cloned age matched controls, however, the authors should mention why these - more molecular - endpoints were not used. Finally, the authors should also state that the ultimate endpoints remain to be measured i.e. health-span and life-span. Otherwise the manuscript deserves to be published without further experiments.

Response: The reviewer is correct to point out that we did not have age and genotype matched controls, which limits assessments of telomere length. However, such analyses in live animals are also restricted to specific, easily accessible cell types (e.g. peripheral blood leukocytes) which have limited informative value in complex multicellular organisms. We have now added a paragraph towards the end of the manuscript which discusses these issues.

Reviewer #4 (Expert in bone biology and arthritis; Remarks to the Author):

This is a manuscript reports on a unique population, and the conclusion that the cloned animals are metabolically healthy and do not have OA is an important finding.

The manuscript reports on a novel animal population, and as such the findings are novel.

The manuscript is primarily descriptive, and as such could be misleading in its conclusion. Since a small number of animals underwent a large number of analyses, from a statistical standpoint drawing any conclusions are problematic. The study likely does not have the power to detect anything but the most drastic differences when so many parameters are analyzed. As such, the conclusions and results should be modified to also include what the power of this study would have to detect each parameter using a Bonferroni correction for multiple variables.

Response: We agree with the reviewer that the study population was small compared to some studies undertaken with regular sheep. The scale of this study, however, was similar to others working with SCNT clones in large animal species. The reviewer's concerns relate to Type I and II errors. That is, on the one hand the reviewer's concern's relate to inferences based on multiple variables which could indicate by chance more differences than actually exist (Type I error), whilst on the other hand they relate to inferences based on experiments which could be perceived as being underpowered (i.e. too few animals per treatment group to detect real effects (Type II error)). However, significant effects were detected for some parameters and below we discuss variance and experimental power. The fact that these effects may be chance effects are not really an issue in this study as the levels reported for each parameter lay within normal reference ranges. However, we have re-run our analyses for each parameter and added Bonferroni correction for comparisons between means, and have modified the data in the text and figures accordingly. Not surprisingly, these adjustments haven't altered our conclusions.

We had been cautious in making inferences in this study and made constant reference to expected normal values (ranges) from published data sets to define clinical significance. However, quantitative analyses were based on three study groups, i.e. 11 ET-Controls, 6 Lleyn and 4 Finn-Dorset clones. The power of our experiments came from the precision afforded by the manner in which they were conducted (e.g. individualised feeding, standardising body fat level (where possible), uniformity in timing of measurements and repeated measurements), and the fact that the two smaller groups were comprised of cloned sheep (with inherently low between-animal variation).

We can illustrate, retrospectively, the 'power' of these experiments with three examples based on resting blood glucose (in fed animals prior to glucose tolerance test), resting systolic blood pressure, and heart rate, together with reference to the following article:

WE Berndston (1991). A simple, rapid and reliable method for selecting or assessing the number of replicates for animal experimentation. *J Anim Sci* **69**, 67-76.

For the 4 Finn-Dorset clones in our study:

a. Plasma glucose: Coefficient of variation (CV) = 4%, therefore 80% chance of detecting a 10% difference at $P < 0.05$ or a 90% chance of detecting 15% difference at $P < 0.05$. Upper 95% Confidence Interval = 4.01 mmol/L, which is well within the normal range (3.25 to 4.50 mmol/L) for fasted (i.e. not fed as in our study) sheep as stated on Line 88 (Page 3) of manuscript. Bearing in mind that 'n' for ET-Controls was 11 and for Lleyn clones was 6 the experiment was powerful enough to detect very modest differences between groups. However, as all our values lay within the normal reference range, differences reported were of no clinical significance.

b. Systolic blood pressure: CV = 6%, 80% chance of detecting a 15% difference at $P < 0.05$ or a 90% chance of detecting a 20% difference at $P < 0.05$. Upper 95% Confidence Interval of 141 mm Hg for Finn-Dorset clones and 137 mm Hg for ET-Controls. In clinical studies, humans with systolic

blood pressures above 140 mm Hg are considered hypertensive. Corresponding data for sheep do not exist. Differences in systolic blood pressure between Finn-Dorset clones and ET-Controls were small (differences in mean systolic blood pressure = 1 mm Hg (i.e. 125 vs 124 mm Hg); Upper 95% Confidence Interval = 4 mm Hg (see above)) and therefore clinically insignificant.

c. Heart rate: CV = 10%, 80% chance of detecting a 25% at $P < 0.05$ or a 90% chance of detecting a 30% difference at $P < 0.05$. In the current study we report higher ($P < 0.001$) heart rates for cloned vs ET-Control sheep (91 vs 76 beats/min). This represents a 20% increase. Again we must bear in mind group sizes of 4, 6 and 11 sheep, The CV above is based on 4 Finn-Dorset clones and the fact that we report differences demonstrates that there was sufficient 'power' in this study.

While the use of a standard diet is useful in glucose tolerance analysis, it is not clear what the role in OA and bone health might be.

Response: In general, the use of a standard diet represents good experimental practice. Diets used in the current study were formulated to meet animal requirements for calcium and phosphorus as well as micro-minerals, thus eliminating nutritional (environmental) effects on bone development between experimental groups. Furthermore, as discussed in the manuscript (Page 8), because animal weight and body composition can influence the clinical manifestation of OA it was important to standardise these variables as far as possible.

The detailed quantitative analysis of the radiographs and MRI analysis should be undertaken using measures taken by multiple individuals with radiographic analysis expertise blinded to the source of the animals. OARSI has developed standards for such measures that should be used, and these include cartilage volume for the MRI studies, and joint space analysis on the radiographs.

Response: We now include formal radiographic assessments from a second independent expert in veterinary orthopaedics who conducted their analyses with no previous knowledge of the study, and they were unaware of animal identity. Using the internationally recognised 4-point scale adopted in the current study, quantitative assessments would require a large number of assessors to provide statistical significance. Our original assessment was very stringent in our rating of lesions which is now confirmed by the lower scores awarded by the second reviewer, whose assessments nevertheless paralleled those of Reviewer 1. Each reviewer independently identified clone 2262 with a higher mineralisation of the right stifle. Consequently, we can see no scientific gain in seeking further independent evaluations.

Joint space analysis is only relevant to weight-bearing animals. In the current study animals were sedated and recumbent during assessment. In addition, as only one of the 4 Finn-Dorset clones showed radiographic evidence of mild to moderate OA we conducted MRI only on the stifle joints of these 4 animals. MRI confirmed diagnoses for that animal. Further quantitative analyses of these data would in our opinion be of no scientific benefit.

Reviewers' Comments:

Reviewer #4 (Remarks to the Author)

All of the issues raised have been adequately addressed in this revision.